# Effect of Heat-Moisture Treatments on Digestibility and Physicochemical Property of Whole Quinoa Flour

**DOI:** 10.3390/foods10123042

**Published:** 2021-12-08

**Authors:** Jilin Dong, Lu Huang, Wenwen Chen, Yingying Zhu, Baoqing Dun, Ruiling Shen

**Affiliations:** 1College of Food and Bioengineering, Zhengzhou University of Light Industry, Zhengzhou 450002, China; djl1968@163.com (J.D.); huanglu3479@163.com (L.H.); chenwenwen1112@163.com (W.C.); zhuying881020@163.com (Y.Z.); 2Henan Key Laboratory of Cold Chain Food Quality and Safety Control, Zhengzhou University of Light Industry, Zhengzhou 450002, China; 3Collaborative Innovation Center of Food Production and Safety, Zhengzhou University of Light Industry, Zhengzhou 450002, China; 4Institute of Crop Science, Chinese Academy of Agricultural Sciences, Beijing 100081, China

**Keywords:** quinoa, flour, heat-moisture treatment, digestibility, physico-chemical property

## Abstract

The starch digestion processing of whole grain foods is associated with its health benefits in improving insulin resistance. This study modified the digestibility of whole quinoa flour (WQ) via heat-moisture treatment (HMT), HMT combined with pullulanase (HMT+P), HMT combined with microwave (HMT+M), and HMT combined with citric acids (HMT+A), respectively. Results showed that all the treatments significantly increased (*p* < 0.05) the total dietary fiber (TDF) content, amylose content, and resistant starch (RS) content, however, significantly decreased (*p* < 0.05) the amylopectin content and rapidly digestible starch (RDS) content of WQ. HMT+P brought the highest TDF content (15.3%), amylose content (31.24%), and RS content (15.71%), and the lowest amylopecyin content (30.02%) and RDS content (23.65%). HMT+M brought the highest slowly digestible starch (SDS) content (25.09%). The estimated glycemic index (eGI) was respectively reduced from 74.36 to 70.59, 65.87, 69.79, and 69.12 by HMT, HMT+P, HMT+M, and HMT+A. Moreover, a significant and consistent reduction in the heat enthalpy (ΔH) of WQ was observed (*p* < 0.05), after four treatments. All these effects were caused by changes in the starch structure, as evidenced by the observed conjunction of protein and starch by a confocal laser scanning microscope (CLSM), the decrease in relative crystallinity, and transformation of starch crystal.

## 1. Introduction

Global statistics indicate that diabetes mellitus (DM) has reached the epidemic level around the world, and one-tenth of people will suffer from this condition by 2030 [1]. In recent years, whole grain foods have received widespread attention because of the benefits in preventing diabetes and improving insulin resistance [2,3]. The antidiabetic effect of whole grain foods is mainly due to the slower starch digestion rate and lower glycemic index (GI), which is beneficial in stabilizing postprandial blood glucose [4]. It was reported that various modification methods can affect the starch digestion process, and among them, heat-moisture treatment (HMT) has received increasing attention due to its features of high efficiency, safety, and environmental protection [5]. HMT is carried out under restricted moisture level range of 10–30% at a temperature of 90–120 °C for 0.5–16 h [6]. A series of previous studies reported that HMT reduced the digestion rate and GI value of starch through increasing the slowly digestible starch (SDS) content and resistant starch content (RS), while decreasing the rapidly digestible starch (RDS) content [7,8]. HMT was also found to change the physicochemical property of starch, such as decreasing the relative crystallinity, changing the starch type from B to A, decreasing the swelling capacity and solubility, increasing the phase transition temperature, and widening the gelatinization temperature ranges [9]. Moreover, other treatments, such as physical, chemical, and enzymatic modifications, are usually combined with HMT to enhance the modification effects. The RS content of corn starch treated by acids-HMT reached 90%, and the GI value reduced from 74 to 49.7 [10]. Cheng et al. found that parboiling-HMT resulted in the formation of rice amylose lipid complexes, and the SDS and RS content increased to 45.8% and 10.4%, respectively [11].

Based on the modification effects on starch, HMT has gradually been used to modify the digestibility and physicochemical property of whole grains. Zheng et al. found HMT decreased the digestion rate of whole highland barley flour and enhanced its health benefits which include reduction in the body weight and serum glucose, improving oxidation resistance and altering the composition of gut microbiota in rats [12]. Consistent with the effects on starch, HMT significantly changed the physicochemical property of whole grains. Xiao et al. reported that HMT significantly reduced the enthalpy of tartary buckwheat flour [13]. Chen et al. found that HMT reduced the peak viscosity of whole wheat flour and increased the gelatinization temperature [14]. Moreover, the formation of starch–lipid complexes, the agglomeration of starch granules, and denature of proteins were observed during HMT. Quinoa (*Chenopodium quinoa Willd*.) has recently attracted increasing attention due to its excellent nutritional compositions [15]. Whole quinoa was reported to have slower in vitro digestion rate, lower estimated glycemic index (eGI), and higher total dietary fiber content than wheat flour [16]. A series of studies showed that consumption of quinoa served to prevent diabetes [17,18]. Thus, whole quinoa flour has been more and more used in production of functional foods that have anti-diabetic effects [19]. If HMT or other treatments combined with HMT can further slow the digestion rate and reduce the GI value of whole quinoa, it would be more beneficial to develop quinoa-based functional foods. However, at present, limited research has focused on the effect of different modification methods on the digestibility and physicochemical property of whole quinoa flour (WQ).

Therefore, the major purpose of the present study is to investigate the modification effects of HMT and HMT combined with other treatments on WQ. The nutritional characteristics, digestibility, physicochemical properties, and granule morphology were respectively assessed.

## 2. Materials and Methods

### 2.1. Raw Materials and Enzyme Solution

Three colored (red:black:white = 1:1:1) whole quinoa flour (WQ) produced in the year of 2019 were purchased from Sanjiang Fertile Soil Co., Ltd. (Qinghai, China). Total starch assay, glucose oxidase/peroxidase (GOPOD), dietary fiber, and β-glucan assay kits were provided by Megazyme International Ireland Ltd. (Bray, Ireland). Pullulanase (EC 3.2.1.41), amyloglucosidase (EC 3.2.1.3), α-amylase (EC 3.2.1.1), pepsin (EC 3.4.23.1), and trypsin (EC 3.4.4.4) were provided by Sigma Chemical Co. (St. Louis, MO, USA). Other chemical reagents obtained the standards of the analysis grade.

Hydrochloric acid-potassium chloride with pH = 2.0 was used to dissolve pepsin (10.0 mg/mL), then the miscible liquid was stored at 4 °C. Suitable amounts of α-amylase and amyloglucosidase were dissolved in 0.2 M sodium acetate buffer (pH = 5.6, mixed with 40 mM calcium chloride), and the concentrations were 290 and 60 U/mL, respectively.

### 2.2. Preparation of Samples

WQ was heat-moisture treated alone (HMT) or with pullulanase (HMT+P), microwave (HMT+M), and citric acids (HMT+A), respectively, according to the conditions obtained from previous optimal experiments (unpublished data). Briefly, the moisture content of WQ was adjusted to 30%, and then was sealed in a plastic bag. After equilibrium at 4 °C for 24 h, WQ were transferred to a steel pan and heated in an oven at 110 °C for 90 min, and then dried at 45 °C, and milled through an 80-mesh sieve. Then, for HMT+P treatment, HMT treated WQ was suspended in phosphate buffer solution (PBS, 0.1 M, pH 5.8) at 55 °C Pullulanase (40 U/g flour) was added to the slurry. After reaction for 8 h, the enzyme was inactivated by boiling water, and the pH value was adjusted to 6.8. The slurry was dried at 45 °C and milled through an 80-mesh sieve. For HMT+M treatment, HMT treated WQ was suspended in distilled water (1:10, w/v), and placed into a microwave oven (600 W, 5 min, G70D20CN1P-D2 Microwave oven, Guangdong Galanz Enterprises Co., Ltd., Foshan, China). The treated sample was dried at 45 °C and milled through an 80-mesh sieve. For HMT+A, HMT treated WQ was suspended in citric acids solution (0.2 M, 1:10, w/v), and kept in a 50 °C water bath for 45 min. Then, the pH value was adjusted to 6.8. The slurry was dried at 45 °C and milled through an 80-mesh sieve.

### 2.3. Nutrient Composition Analysis

The crude protein (method 46–12), crude fat (method 30–25), and ash (method 8–01) contents of samples were determined using AACC (2000) standard methods. Total dietary fibers (TDF) were determined using the dietary fiber kit (Megazyme International, Bray, Ireland) following the approved AOAC 985.29. Amylose/amylopectin content was determined using K-AMYL 12/16. AOAC 996.11 was used to determine the total starch content.

### 2.4. Digestibility In Vitro

The method of Yu et al., (2017) was used to determine the extent of in vitro starch hydrolysis [20]. In total, 6 mL deionized water and samples (containing 100 mg starch, dry weight basis) were mixed together, then the solution was mixed with 2 mL pepsin solution (37 °C shaking for 30 min). The pH of suspensions was modified to 7, 3 mL amylase-amyloglucosidase solution were added into the above solution, shaken for 180 min at 37 °C. At special time intervals (0, 20, 40, 60, 80, 100, 120, 140, 160, 180 min), to inactivate the enzymes, 1 mL aliquots and 9 mL of 0.3 M Na_2_CO_3_ solution were added. After centrifugation, 3,5-dinitrosali-cyclic acid assay was used to determine the reducing sugar content of supernatant. Hydrolysisrate, RDS content, SDS content, and RS content was calculated as follows:Hydrolysisrate (%) = reducing sugar content/total starch × 100(1)
RDS (%) = (G_20_ − FG)/TS × 100(2)
SDS (%) = ((G_120_ − G_20_) × 0.9)/TS × 100(3)
RS (%) = (TS − RDS − SDS)/TS × 100(4)
where: FG was the content of free reducing sugar in the flour before enzymatic hydrolysis (mg); TS was the total starch content (mg); G_20_ was the reducing sugar content (mg) produced after 20 min of enzymatic hydrolysis; G_120_ was the reducing sugar content (mg) produced after 120 min of enzymatic hydrolysis.

The GI were analyzed with a modified procedure established by Goni et al. [21]. The starch hydrolysis curves follow a first order equation:C = C∞ (1 − e^−kt^)(5)
where C is the concentration at t time; C∞ is the equilibrium concentration, k is the kinetic constant.

The area under the hydrolysis curve (AUC) can be calculated as follows:AUC = C∞ (t_f_ − t_0_) − (C∞/k) (1 − exp(−k(t_f_ − t_0_)))(6)
where t_f_ is the final reaction time (180 min) and t_0_ is the initial reaction time (0 min).

The hydrolysis index (HI) is the ratio of AUC between each sample and that of white bread (7300). The eGI was calculated using the equation:eGI = 39.71 + 0.549 × HI(7)

The experiments were carried out at least in triplicate to achieve re-producible results.

### 2.5. Thermal Property

A Differential Scanning Calorimeter (Q100, TA Instruments, New Castle, DE, USA) was used to analyze the samples’ thermal properties and the result was described by the mothed of Huang, Dong, Zhang, Zhu, and Qu with some modifications [22]. Six microliters of distilled water and 3 mg (dry weight basis) sample were mixed together and placed in sealed aluminum pans. Before heating from 30 to 150 °C with a rate of 10 °C/min, the pans were left for 24 h at room temperature. To calibrate the DSC instrument, an empty aluminum pan was used as the reference. All measurements were performed in triplicate.

### 2.6. Crystalline Analysis

The method of Jan et al. was used to extract quinoa starch with some modifications [23]. The quinoa and n-pentane were mixed uniformly at a ratio of 1:5 and leached at 50 °C for 3 h. After centrifugation at 2683 × g for 10 min, the precipitate and 0.2% NaOH solution were stirred for 3 h at a ratio of 1:5. Then, the mixture was passed through a 200-meshes sieve, and the undersize was centrifuged at 4000 r/min for 10 min. The bottom white material was washed with 5% NaCl solution and distilled water 3 times, respectively. The upper gray material was scraped off, and the white substance at below was put into a 40 °C oven to dry for 24 h, which was crude starch. The crystallinity degree and polymorphic composition of starch samples were obtained using an X-ray diffractometer (PW-1710, Philips, The Netherlands). The characteristic ray was a Cu-Ka source (λ = 0.154 nm) operating at 40 kV and 30 mA, respectively. The samples were examined over an angular range of 5–40° (2θ) at a scanning speed of 2°/min and a step size of 0.02°. All measurements were performed in triplicate. The relative crystallinity (%) was estimated with MDI-Jade 6.5 software (Material Date, Inc.Livermore, California, USA) based on the following equation:Relative crystallinity = A0/At × 100%(8)

A0 denotes the area under each peak, and At means the total area under the diffractograms.

### 2.7. Granule Morphology

The morphology of the starch granules was examined by a scanning electron microscope (SEM, JSM-6490LV FEI Instruments, Hillsboro, OR, USA) and confocal laser scanning microscopy (CLSM, LSM710, Carl Zeiss Inc., Germany) was respectively used to examine the morphology of the starch granules as described previously [24,25]. Samples were freeze-dried before observation. The cross-section and surfaces of the samples were sputtered with gold at an acceleration voltage of 20 kV, before being observed and photographed. Samples (1 g) were immersed in deionized water (5 mL) for 4 h and dyed (about 30 mg) using the mixed solution with fluorescein isothiocyanate (FITC, 0.2 mg/mL) and rhodamine B (0.02 mg/mL) at a ratio of 7:3. All samples were placed under dark conditions for a short time to avoid fluorescence quenching. All images were analyzed using ZEN Imaging software. Parameter settings: FITC, fluorescence excitation wavelength was 488 nm, fluorescence emission wavelength was 450–540 nm; Rhodamine B fluorescence excitation wavelength was 543 nm, fluorescence emission wavelength was 545–660 nm. All measurements were performed in triplicate.

### 2.8. Statistical Analysis

Data was expressed as the mean ± SD, included at least 3 replicates per sample. One-way Analysis of Variance (ANOVA) and Tukey’s test were performed Statistical Product and Service Solutions (SPSS) version 22.0 (International Business Machines Corporation, Armonk, NY, USA). Statistical significance was set at *p* < 0.05.

## 3. Results and Discussion

### 3.1. The Effects of Different Treatments on the Nutrient Composition of WQ

The effects of different treatments on WQ nutrients are shown in Table 1. The TDF content increased 4.18%, 15.82%, 9.20%, and 5.70% by HMT, HMT+P, HMT+M, and HMT+A (*p* < 0.05), respectively, which was consistent with a previous report that the TDF content in whole wheat flour significantly increased after HMT [26]. This phenomenon may be due to the formation of RS, which was known to enhance the dietary fiber contents in foods [27,28]. Additionally, the content of amylose was increased by different treatments: HMT+P increased the most, by 31.24% (*p* < 0.05), and HMT increased the least, by 9.25% (*p* < 0.05). The results agreed with the results of Niu et al. who reported that HMT+M increased the amylose content in potato starch [29]. Additionally, the amylopectin content decreased significantly (*p* < 0.05), reduced by 3.12% and 44.53% for HMT and HMT+P, respectively. As previously reported, the increase in amylose content and the reduction in amylopectin content resulted from the breakage of some covalent linkages by excessive heating, and the degradation of the exterior linear chains of amylopectin during HMT [30,31]. The protein, ash, and total starch content had no obvious change (*p* < 0.05). Overall, HMT+P affected nutritional components most among the four HMT methods.

### 3.2. The Effects of Different Treatments on the Digestibility In Vivo of WQ

The RS, SDS, and RDS content and eGI value of WQ under different treatments are shown in Table 2. As shown, HMT changed the starch composition of WQ. The RS content was increased (*p* < 0.05) from 2.89% (WQ) to 5.32% (HMT), 15.71% (HMT+P), 7.74% (HMT+M), and 12.49% (HMT+A), respectively. The results were consistent with Hung et al. who found that the RS content in rice was significantly increased by HMT [32]. Moreover, HMT+P, HMT+M, and HMT+A increased (*p* < 0.05) the SDS content by 19.13%, 36.73%, and 20.27%, respectively. Asranudin et al. found that the SDS content increased to 4–16% in purple yam flour after heat-moisture treatment combined with bake treatment [9]. It was reported that the varying effects on SDS content depend on the source of the starch and modification conditions [7]. In contrast, all treatments reduced the RDS content. The RDS content was decreased, ranging from 5.26% (HMT) to 41.05% (HMT+P). Asranudin et al. obtained a similar result that the RDS content of purple yam flour reduced to 9–17% after heat-moisture treatment combined with bake treatment [9]. These changes may be due to the rearrangement of crystalline structure inside granules and the enhancement of the interaction of both amylose and amylose-amylopectin chains during HMT [33].

In addition, different treatments reduced the eGI value to varying degrees: 74.36 (WQ) > 70.59 (HMT) > 69.79 (HMT+M) > 69.12 (HMT+A) > 65.87 (HMT+P). The results may be due to the changes of starch profile which reduced the starch digestibility. Among all the treatments, HMT+P had the greatest impact on WQ (*p* < 0.05). The higher RS content and the lower RDS content led to the lower eGI value.

### 3.3. The Effects of Different Treatments on the Thermal Property of WQ

The thermal property of different samples is shown in Table 3. Compared with WQ, all modified samples showed higher gelatinization temperature (T_0_), peak gelatinization temperature (T_p_), and conclusion gelatinization temperature (T_c_), but lower ΔH. The gelatinization enthalpy (ΔH) is defined as the energy required for breaking the molecular interactions within the starch structure during gelatinization, mainly reflecting the dissolution of the double helix structure of starch [34].

As shown, all treatments significantly reduced the ΔH (*p* < 0.05) from 4.33 J/g to 1.05 J/g (HMT), 1.07 J/g (HMT+P), 0.53 J/g (HMT+M), and 0.64 J/g (HMT+A), respectively. Consistent with our results, the ΔH of tartary buckwheat flour also reduced from 6.2 to 3.8 J/g after HMT [13]. It was reported that the decreased ΔH was related to the dissociation of double helices in crystalline and the disruption of amorphous regions within starch granules during HMT [35].

### 3.4. The Effects of Different Treatments on the Crystalline Property of WQ

The crystallization of starch granules mainly occurs in the outer chain of amylopectin, and the X-ray diffraction peak of starch is usually used to reflect the characteristics of amylopectin microcrystalline bundles. The X-ray diffraction pattern and relative crystallinity of different treatments of quinoa starch are shown in Figure 1. Quinoa starch exhibited type “A” X-ray pattern, with peaks located at 2θ = 15.22°, 17.12°, 18.04°, and 23.42°. Type A polymorphs are usually formed from amylopectin with shorter amylopectin. The B-type X-ray pattern is typical of root starches, tubers of starchy cereals rich in amylose, and certain fruits, with diffraction peaks of greatest intensity at 5.6°, 14.4°, 17.2°, 22.2°, and 24° in 2θ. The type of crystallinity can appear in starches, V-type, which results from the crystallization of amylose with lipids; this type exhibits peaks of intensity at characteristic diffraction angles of 12.6°, 13.2°, 19.4°, and 20.6° in 2θ. V-shaped components appeared in the particles after HMT. It was consistent with the results of [14], the crystal form of wheat flour also changed from type A to type A+V after HMT due to the better stability of V type components. The HMT+P samples showed a strong peak at 15.22°, 17.12°, 20°, 22°, 24°, indicating HMT+P caused a change in the crystalline arrangement of A-type to A+B+V-type. Type V polymorphs which derived from the single helix complex of amylose and endogenous lipids, were more pronounced in starches with high amylose content [36]. The amorphous region increased, and the peak almost disappeared after HMT+M, and the whole curve was close to the peak of steamed bread, indicating that the crystalline structure of starch was seriously damaged. After HMT+C, the structural damage is more serious, which may be due to the degradation of starch chain by citric acid [19].

HMT not only changed the crystal form of starch granules, but also reduced the relative crystallinity of starch granules. The relative crystallinity was decreased 6.77%, 13.55%, 33.86%, and 52.19% by HMT, HMT+P, HMT+M, and HMT+A, respectively (*p* < 0.05). The results agreed with Delinski et al. who reported that the relative crystallinity of amaranth starch decreased from 32.26% to 28.79% after HMT [37]. It was due to the fact that HMT accelerated the movement and rearrangement of molecular chains, thus forming a more disordered structure, which caused a decrease in the relative crystallinity of starch. Additionally, previous studies have shown that the degree of relative crystallinity decline obviously depends on the amylose content [36]. In addition, the effect on relative crystallinity of HMT+P, HMT+M, and HMT+A was more obvious compared with HMT. Those findings may be because HMT+P destroyed part of the double helix structure [19], HMT+M increased the amount of submicron crystals, and HMT+A degraded part of the starch chain, resulting in a decrease in crystallinity [38]. In summary, HMT+A affected relative crystallinity of the WQ starch most.

### 3.5. The Effects of Different Treatments on the Microstructure of WQ

The microstructure of WQ with different treatments is shown in Figure 2. WQ particle had a polygonal shape with no cracks. HMT caused the surface of the particles to become rough and part of the pits appeared compared to WQ. Additionally, the particles became larger with irregular shapes after HMT. The results were consistent with the research on sorghum and hemp resin starch by HMT [39].

Compared with HMT, HMT+P made the original particle structure and properties disappear, such as the rough and irregular surface and the larger particles. HMT+M made the granular structure disappear, the particles became larger, and the surface became rough compared to HMT. Han et al. believed that in the process of microwave radiation, the surface became rough easily due to heat effect from the inside to outside [40]. Compared with HMT, HMT+A made the particle shape into an irregular lamellar stacked structure with cracks on the surface. Hung et al. obtained similar results; RS of potato exhibited a lamellar stacked structure after the process of HMT combined with organic acid [41]. It may be because the high temperature during HMT evaporated the moisture to the surface, which promoted the adhesion of the particles and made the particles larger; when the process finished, the drop in temperature caused the surface of the particles to collapse and pits to appear.

In order to observe conjunction of protein and starch of different samples, a CLSM was used, and the result is shown in Figure 3. Compared with WQ which was dispersed and complex, HMT promoted the compounding between starch and protein. There were more protein rich spots (indicated by arrows) after treatment. Moreover, the composite degree of HMT+M, HMT+A, and HMT+P was greater than HMT. Consistent with the results of [14], HMT benefited the aggregation between proteins and starch. It may be because HMT caused starch granules agglomeration and protein denaturation, and the denatured proteins dispersed and adhered to the surface of starch granule agglomerates [42].

## 4. Conclusions

In conclusion, the nutritional composition, digestibility, and physicochemical property of WQ were modified by HMT, HMT+P, HMT+M, and HMT+A. HMT+P led to the highest TDF, RS content, and the lowest eGI value. All treatments significantly reduced the ΔH of WQ. The conjunction of protein and starch was observed after four treatments, especially for HMT+M. HMT+A brought the most significant changes in the starch structure, as evidenced by the decrease in relative crystallinity and transformation of starch crystal. Overall, among the four treatments, HMT+P affected nutritional components most among the four HMT methods and HMT+A had the greatest effect on the physical and chemical properties of WQ. The results are expected to provide theoretical support for the development of functional food of quinoa.

## Figures and Tables

**Figure 1 foods-10-03042-f001:**
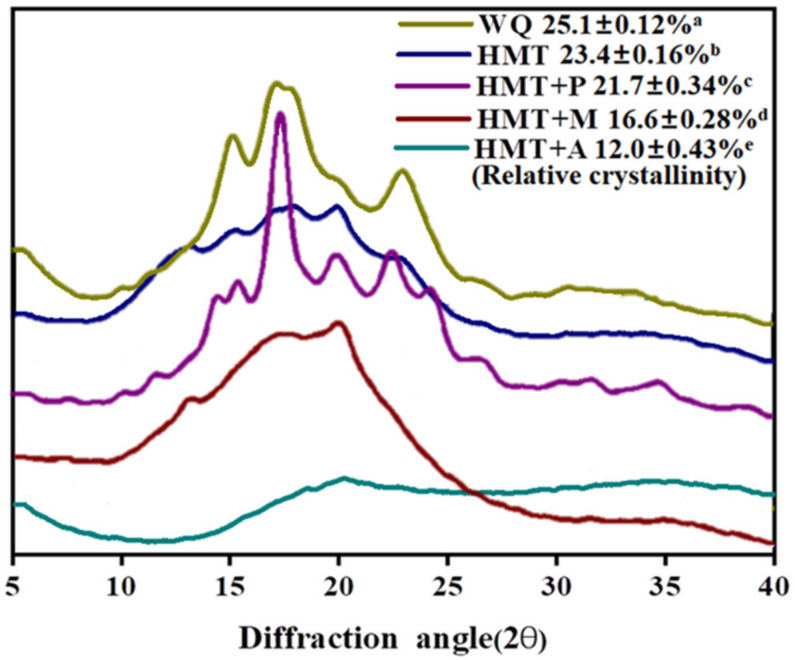
X-ray diffraction pattern and relative crystallinity of starch with different treatments. WQ: whole grain quinoa flour; HMT: heat-moisture treated whole grain quinoa flour; HMT+P: HMT combined with pullulanase treated whole grain quinoa flour; HMT+M: HMT combined with microwave treated whole grain quinoa flour; HMT+A: HMT combined with acids treated whole grain quinoa flour. Different letter are significantly different (*p* < 0.05).

**Figure 2 foods-10-03042-f002:**
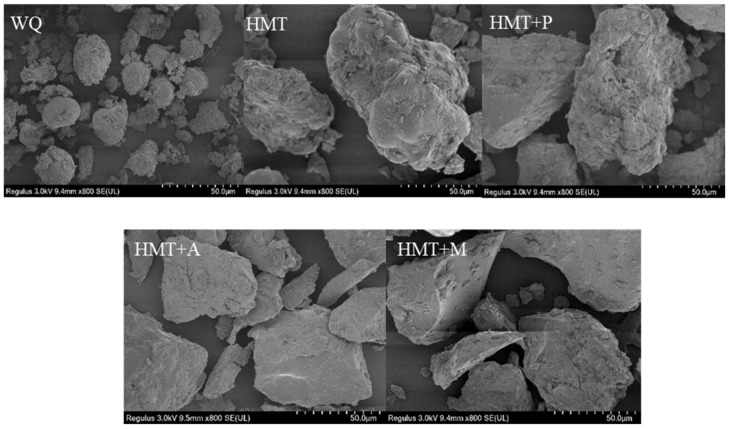
Scanning electron microscope A (×800) of WQ with different treatments. (**WQ**): whole grain quinoa flour; (**HMT**): heat-moisture treated whole grain quinoa flour; (**HMT+P**): HMT combined with pullulanase treated whole grain quinoa flour; (**HMT+M**): HMT combined with microwave treated whole grain quinoa flour; (**HMT+A**): HMT combined with acids treated whole grain quinoa flour.

**Figure 3 foods-10-03042-f003:**
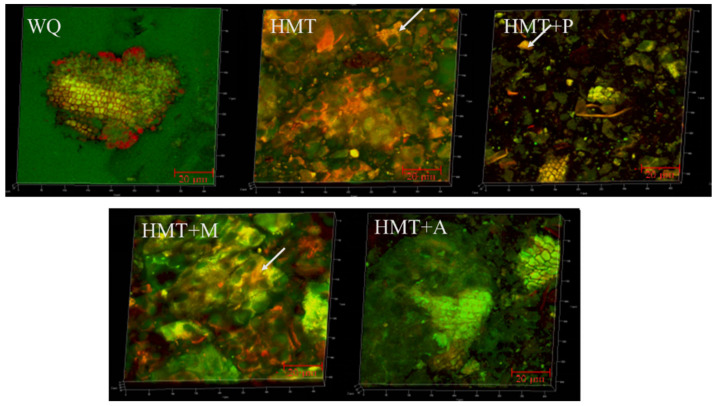
Confocal laser scanning microscope B (×800, red represents protein, green represents starch) of WQ with different treatments. (**WQ**): whole grain quinoa flour; (**HMT**): heat-moisture treated whole grain quinoa flour; (**HMT+P**): HMT combined with pullulanase treated whole grain quinoa flour; (**HMT+M**): HMT combined with microwave treated whole grain quinoa flour; (**HMT+A**): HMT combined with acids treated whole grain quinoa flour.

**Table 1 foods-10-03042-t001:** Effects of different treatments on the nutritional components of WQ (%, dry weight).

Nutrients	WQ	HMT	HMT+P	HMT+M	HMT+A
Protein	14.88 ± 0.93 ^a^	14.70 ± 0.95 ^a^	15.18 ± 0.99 ^a^	15.03 ± 0.96 ^a^	14.39 ± 0.94 ^a^
Total fat	6.83 ± 0.07 ^a^	6.52 ± 0.08 ^b^	6.29 ± 0.10 ^b^	6.34 ± 0.06 ^b^	6.47 ± 0.03 ^b^
Ash	2.48 ± 0.08 ^a^	2.46 ± 0.07 ^a^	2.39 ± 0.10 ^a^	2.47 ± 0.09 ^a^	2.50 ± 0.11 ^a^
Total starch	61.56 ± 1.86 ^a^	61.68 ± 1.88 ^a^	60.76 ± 1.76 ^a^	61.56 ± 1.77 ^a^	61.55 ± 1.74 ^a^
TDF	13.15 ± 0.56 ^d^	13.70 ± 0.48 ^c^	15.23 ± 0.58 ^a^	14.36 ± 0.41 ^b^	13.9 ± 0.52 ^c^
Amylose	7.44 ± 0.38 ^e^	9.25 ± 0.41 ^d^	31.24 ± 0.44 ^a^	22.91 ± 0.39 ^b^	19.86 ± 0.41 ^c^
Amylopectin	54.12 ± 1.10 ^a^	52.43 ± 1.09 ^b^	30.02 ± 1.09 ^e^	38.45 ± 1.10 ^d^	41.69 ± 1.09 ^c^

All the values are reported in dry weight basis. Data was expressed as the mean ± SD, included at least 3 replicates per sample, and followed by a different letter within the column are significantly different (*p* < 0.05). One-way ANOVA and Tukey’s test were performed using SPSS version 22.0 (IBM Corp., Armonk, NY, USA). Statistical significance was set at *p* < 0.05. WQ: whole grain quinoa flour; HMT: heat-moisture treated whole grain quinoa flour; HMT+P: HMT combined with pullulanase treated whole grain quinoa flour; HMT+M: HMT combined with microwave treated whole grain quinoa flour; HMT+A: HMT combined with acids treated whole grain quinoa flour.

**Table 2 foods-10-03042-t002:** Effects of different treatment methods on the RS, SDS, and RDS content and eGI of whole quinoa flour (%, dry weight).

Samples	WQ	HMT	HMT+P	HMT+M	HMT+A
RS	2.89 ± 0.55 ^e^	5.32 ± 0.52 ^d^	15.71 ± 0.56 ^a^	7.74 ± 0.54 ^c^	12.49 ± 0.58 ^b^
SDS	18.54 ± 0.46 ^c^	18.35 ± 0.43 ^c^	21.86 ± 0.40 ^b^	25.09 ± 0.43 ^a^	22.07 ± 0.41 ^b^
RDS	40.12 ± 1.54 ^a^	38.01 ± 1.52 ^a^	23.65 ± 1.51 ^c^	28.43 ± 1.49 ^b^	26.99 ± 1.50 ^b,c^
eGI	74.36 ± 1.96 ^a^	70.59 ± 1.88 ^b^	65.87 ± 1.76 ^c^	69.79 ± 1.87 ^b^	69.12 ± 1.56 ^b^

All the values are reported in dry weight basis. Data was expressed as the mean ± SD, included at least 3 replicates per sample, and followed by a different letter within the column are significantly different (*p* < 0.05). One-way ANOVA and Tukey’s test were performed using SPSS version 22.0 (IBM Corp., Armonk, NY, USA). Statistical significance was set at *p* < 0.05. WQ: whole grain quinoa flour; HMT: heat-moisture treated whole grain quinoa flour; HMT+P: HMT combined with pullulanase treated whole grain quinoa flour; HMT+M: HMT combined with microwave treated whole grain quinoa flour; HMT+A: HMT combined with acids treated whole grain quinoa flour.

**Table 3 foods-10-03042-t003:** Effects of different treatment methods on the RS, SDS, and RDS content and eGI of whole quinoa flour (%, dry weight).

Samples	T_0_	T_p_	T_c_	△H (J/g)
WQ	65.05 ± 0.37 ^e^	72.78 ± 0.42 ^e^	81.09 ± 0.35 ^e^	4.33 ± 0.003 ^a^
HMT	97.84 ± 0.46 ^d^	107.84 ± 0.53 ^d^	117.86 ± 0.70 ^d^	1.05 ± 0.001 ^b^
HMT+P	100.75 ± 0.74 ^c^	109.91 ± 0.43 ^c^	117.35 ± 0.61 ^c^	1.07 ± 0.006 ^b^
HMT+M	104.06 ± 0.53 ^b^	112.77 ± 0.67 ^b^	124.37 ± 0.53 ^b^	0.53 ± 0.002 ^c^
HMT+A	113.75 ± 0.93 ^a^	115.54 ± 0.73 ^a^	132.67 ± 0.65 ^a^	0.64 ± 0.008 ^c^

All the values are reported in dry weight basis. Data was expressed as the mean ± SD, included at least 3 replicates per sample, and followed by a different letter within the column are significantly different (*p* < 0.05). One-way ANOVA and Tukey’s test were performed using SPSS version 22.0 (IBM Corp., Armonk, NY, USA). Statistical significance was set at *p* < 0.05. WQ: whole grain quinoa flour; HMT: heat-moisture treated whole grain quinoa flour; HMT+P: HMT combined with pullulanase treated whole grain quinoa flour; HMT+M: HMT combined with microwave treated whole grain quinoa flour; HMT+A: HMT combined with acids treated whole grain quinoa flour.

## Data Availability

All data are provided in the manuscript.

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
