# Peer review of "Effect of Heat-Moisture Treatments on Digestibility and Physicochemical Property of Whole Quinoa Flour"

_foods, 2021, doi:10.3390/foods10123042_

Round 1
Reviewer 1 Report
Dear Authors,
in the answer sheet to the questions / recommendations in the first point there is an unfinished sentence; I don't know what the author wanted to write, please fill in and send me the answer again
1. Chapter 3.4 Pasting properties: because the viscosity curves of the modified starch have very low values and can not see differences between them, I recommend show a parameters of gelatinization in the table and eventually curves show in supplementary files. After that can discuss differences between results for starch with method modifications.
A: The Pasting properties was deleted. Because,???
Reviewer 2 Report
I already had the pleasure of judging this article. Previously, I expressed myself very high about him. re-maintains its self-esteem.
A well-written introduction is exhaustive to understanding the research undertaken. The purpose of the work is clearly formulated. Well-described and selected research methods. presentation of the results and their discussion at a high level. The authors do not limit themselves only to the discussion and analysis of one research method, but show the importance of the discussed results for the next (for example, changes in the proportion of amylopectin from the nose with the viscosity of the systems, which is explained by a decrease in the proportion of amylopectin, which is characterized by the cross-linking nature of the system). Very valuable work and information for people struggling not only with a diet with a low glycemic index, but also with gluten intolerance. requests approval of the article in this form.
The introduced changes to the method of sample preparation and more detailed descriptions of the preparation and conduct of determinations, are so precise that a potential reader-researcher can easily repeat them and confront the results with the present ones.
Author Response
Thanks for your comments which help us to improve this manuscript.
Reviewer 3 Report
Detailed recommendation:
Abstract: please add more data to the abstract.
Introduction: please give more information about resistant starch and fiber properties
Key words: add: flour
From which year did you take quinoa seeds?
Did you examined in DSC analysis retrogradation process?
Conclusions: Which product is the best?
Round 2
Reviewer 1 Report
Dear Authors, I accept the manuscript in the present form.
Kind regards
G.A.
This manuscript is a resubmission of an earlier submission. The following is a list of the peer review reports and author responses from that submission.
Round 1
Reviewer 1 Report
This paper shows the influence of heat moisture treatment (HMT) combined with other modification methods on the digestability and physicochemical properties of whole quinoa flour. The layout of the article is generally good. The English language, although generally good, could be improved at some places throughout the article. However, I think the paper requires significant improvement in the methodology and discussion parts. Therefore, I suggest major revision.
Materials and methods:
- Why three color (red: black: white = 1:1:1) whole quinoa flour (WQ) were used for this study? Why not single color whole quinoa flour? What’s the differences in nutritional compositions among the different color whole quinoa flour?
- Lines 93-100, the methodology for sample preparation is not clear. Please elaborate detailed modification conditions, particularly for HMT with pullulanase (HMT+P), microwave (HMT+M), and citric acids (HMT+A), including samples size for each batch.
- Lines 126-130, the authors should explain the time interval they used for hydrolysis analysis. How the area under curve was calculated? How the reference sample (fresh white bread) was prepared for hydrolysis experiment?
- Line 152, centrifugation conditions should be expressed as gravitational force instead of rpm because the centrifugation force depends on the rotor used.
Results and discussion:
- Lines 187-191, the authors consider the TDF content in whole wheat flour was significantly increased after modification due to the formation of RS. However, for TDS, HMT+P> HMT+M> HMT+A >HMT; for RS, HMT+P > HMT+A > HMT+M>HMT. Why the order of increasing in TDF was not consistent with that in RS?
- I suggest that the authors make a better presentation and discussion of the thermal properties by DSC and pasting properties by RVA parts. There were essentially no significant differences among the four HMT methods they studied, however, one would expect that the increase in RS should have some impact on the thermal and pasting properties.
- Figure 2, why the gelatinization curve of WQ showed 2 peaks? Please explain.
- I suggest the authors also prepare a table about the RVA parameters of WQ with various HMT treatment. It seemed that the onset temperature of pasting was retarded by various HMT treatments. The RVA parameters could better differentiate the significance among treatments.
- The authors should also make a better presentation and discussion about the X-ray diffraction results. Description about B type and V type X ray pattern should also be explained, respectively. The changes in structural characteristics should be discussed or linked to the physicochemical properties as well.
- Figure 3 should be modified for clearly reading. Specifically, the sequence of the colored curves should be consistent with the sequence of curve legends.
- Please improve the resolution of Figure 5. It is hard to tell the scale bar and the protein riched spots (indicated by arrows) after treatment in its present form.
Reviewer 2 Report
Some specific comments:
- Line 93-100
- How to proceed to the next stage of processing after WQ is treated by HMT? Are there any recover processes, such as drying and powdering? How to add trace amounts of enzyme and organic acid to HMT WQ homogeneously?
- What is the moisture content of the sample during the second stage (enzyme, microwave and organic acid) of processing?
- What does 0.2M citric acid mean? Is 0.2M citric acid added? What is the amount added?
- How to recover WQ after processing?
- Line 122: “reducing sugar in the starch” should be “reducing sugar in the flour” in this study.
- Line 150~165: Although the analysis of crystallinity requires higher purity starch to obtain better results, the extraction process will also cause part of the sample to be lost and the measurement will lose its representativeness. Especially WQ has been partially gelatinized after HMT treatment, so the result of this analysis is not representative.
- Line 192-196: How does HMT increase the amylose content of starch in WQ? What is the relationship between the amylose content and the content of amorphous and crystalline regions?
- Line 212: What starch composition has changed?
- Section 3.2: The differences in the effects of the four HMT treatments on WQ digestibility and their causes need to be explained in detail
- Section 3.3 & Fig. 1: The range of the Y axis in Figure 1 is too wide to see the difference in heat flow between samples. It is recommended to provide the melting temperatures instead of Figure 1.
- Line 255-257: Related assumptions need more data to support, because this study only uses one sample.
- 3: How to calculate the crystallinity of HMT, HMT+A and HMT+M?
- Line 309 & Fig. 4: According to Figure 4, the surface of WQ is not smooth.
- Line 339 & Fig. 5: No arrows are provided in Figure 5.
Reviewer 3 Report
Recommendation: Reconsider after major revision
Manuscript number: foods-1380452
Article Type: Article
Title: Effect of Heat-moisture Treatments on Digestibility and Physicochemical Property of Whole Quinoa Flour
The Authors have investigated the modification effect of HMT and HMT combined with other treatments on quinoa flour. They have studied nutritional characteristics (crude protein, crude fat, ash, total dietary fibers, amylose/amylopectin content, total starch content), digestibility (in vitro), physicochemical properties (pasting properties, crystalline analysis) and granule morphology.
Comments:
Chapter 3.4 Pasting properties: because the viscosity curves of the modified starch have very low values and can not see differences between them, I recommend show a parameters of gelatinization in the table and eventually curves show in supplementary files. After that can discuss differences between results for starch with method modifications.
Conclusions: lines 346 – 347 – I don’t understand the sentence.
Conclusions: please show in the conclusions which sort of the modification methods had the greatest impact on the WQ properties
Reviewer 4 Report
A well-written introduction is exhaustive to understanding the research undertaken. The purpose of the work is clearly formulated. Well-described and selected research methods. presentation of the results and their discussion at a high level. The authors do not limit themselves only to the discussion and analysis of one research method, but show the importance of the discussed results for the next (for example, changes in the proportion of amylopectin from the nose with the viscosity of the systems, which is explained by a decrease in the proportion of amylopectin, which is characterized by the cross-linking nature of the system). Very valuable work and information for people struggling not only with a diet with a low glycemic index, but also with gluten intolerance. requests approval of the article in this form.